# Tumor Microenvironment-Derived Metabolites: A Guide to Find New Metabolic Therapeutic Targets and Biomarkers

**DOI:** 10.3390/cancers13133230

**Published:** 2021-06-28

**Authors:** Juan C. García-Cañaveras, Agustín Lahoz

**Affiliations:** 1Biomarkers and Precision Medicine Unit, Medical Research Institute-Hospital La Fe, Av. Fernando Abril Martorell 106, 46026 Valencia, Spain; 2Analytical Unit, Medical Research Institute-Hospital La Fe, Av. Fernando Abril Martorell 106, 46026 Valencia, Spain

**Keywords:** biomarkers, cancer metabolism, oncometabolites, metabolomics, imaging, LC-MS, cancer therapy, metabolic inhibitors, MRS, MRI

## Abstract

**Simple Summary:**

Cancer cells reprogram their metabolism to meet the bioenergetic, biosynthetic and redox demands required to maintain tumor formation, growth and dissemination. Additionally, rewired metabolism in the tumor microenvironment contributes to immune evasion by depleting key nutrients required for mounting a proper immune response, but also by producing immunosuppressive metabolites. Altered cancer metabolism can be exploited therapeutically by targeting cancer cell activities required for biomass and energy production, but also by alleviating the immunosuppressive properties of the tumor microenvironment. Using imaging techniques (i.e., magnetic resonance spectroscopy (MRS), positron emission tomography (PET), magnetic resonance imaging (MRI)) or the liquid chromatography coupled to mass spectrometry (LC-MS)-based analysis of biofluids, altered metabolites produced by dysregulated cancer metabolism can be used as noninvasive biomarkers for diagnosis and therapy.

**Abstract:**

Metabolic reprogramming is a hallmark of cancer that enables cancer cells to grow, proliferate and survive. This metabolic rewiring is intrinsically regulated by mutations in oncogenes and tumor suppressors, but also extrinsically by tumor microenvironment factors (nutrient and oxygen availability, cell-to-cell interactions, cytokines, hormones, etc.). Intriguingly, only a few cancers are driven by mutations in metabolic genes, which lead metabolites with oncogenic properties (i.e., oncometabolites) to accumulate. In the last decade, there has been rekindled interest in understanding how dysregulated metabolism and its crosstalk with various cell types in the tumor microenvironment not only sustains biosynthesis and energy production for cancer cells, but also contributes to immune escape. An assessment of dysregulated intratumor metabolism has long since been exploited for cancer diagnosis, monitoring and therapy, as exemplified by 18F-2-deoxyglucose positron emission tomography imaging. However, the efficient delivery of precision medicine demands less invasive, cheaper and faster technologies to precisely predict and monitor therapy response. The metabolomic analysis of tumor and/or microenvironment-derived metabolites in readily accessible biological samples is likely to play an important role in this sense. Here, we review altered cancer metabolism and its crosstalk with the tumor microenvironment to focus on energy and biomass sources, oncometabolites and the production of immunosuppressive metabolites. We provide an overview of current pharmacological approaches targeting such dysregulated metabolic landscapes and noninvasive approaches to characterize cancer metabolism for diagnosis, therapy and efficacy assessment.

## 1. Introduction

Despite the first observations of metabolic alterations in tumors, such as cancer cell dependency on aerobic glycolysis, being first documented one century ago [1], cancer metabolism re-emerged as a hot topic in the field of oncology in the last decade. Presently, tumor-altered metabolism is unquestionably considered a hallmark of cancer [2]. Cancer cells reprogram their metabolism to meet the bioenergetic, biosynthetic and redox demands required to maintain tumor formation, growth and dissemination [3,4,5]. Rather than conducted by mutations in metabolic genes, this metabolic reprogramming is usually intrinsically modulated by the action of oncogenes and suppressor genes (e.g., *KRAS*, *EGFR*, *p53*, *MYC*, *PTEN* or *LKB1*). However, mutations in the metabolic enzymes isocitrate dehydrogenase 1/2 (*IDH1/2*), fumarate hydratase (*FH*) and succinate dehydrogenase (*SDH*) also result in oncogenic transformation through mechanisms exerted by the accumulation of oncometabolites D-2hydroxyglutarate (D-2HG), fumarate and succinate, respectively [3,5].

Established cancers are surrounded by a large heterogeneous collection of infiltrating and resident host cells (fibroblasts, immune cells, adipocytes, endothelial cells), secreted factors and extracellular matrix that are collectively known as the tumor microenvironment (TME) [6,7]. In the TME, these various cells types establish crosstalk mediated not only by direct cell–cell interactions or paracrine signaling (i.e., cytokines, chemokines, growth factors, inflammatory mediators, etc.), but also by metabolic factors (limited nutrient availability, nutrient competition, production of immunomodulatory/signaling metabolites, etc.) [4,5,8]. From a metabolic perspective, such interactions may enhance the activity of a given metabolic enzyme or up-regulate transcription factors that, in turn, increase the expression of metabolic regulators [5] and alters metabolic and/or signaling pathways. The complex interplays in TME results in a dynamic scenario in which metabolic and non-metabolic conditions vary over time. From a biomass and energy production perspective, cancer cells have demonstrated possessing a high plasticity in adapting their metabolism to these changing conditions that enables them to overcome nutrient and oxygen limitations. Some of the enhanced tumor metabolic pathways include up-regulated glycolysis, glutaminolysis, fatty acid (FA) catabolism and OXPHOS, depending on oncogenic mutations, tissue of origin and tissue location [9,10]. In addition to the more efficient use of canonical nutrients and metabolic pathways, cancer cells can exploit diverse carbon and nitrogen sources for biomass and energy production (e.g., branched-chain amino acids (BCAAs), acetate, ammonia, etc.), and even modulate the metabolism of cancer-supporting cells to hijack the key nutrients they need [3,4,5,8,11]. In addition to this, upon the inhibition of a particular enzyme/route, cancer cells can up-regulate compensatory pathways to overcome the targeting of a particular metabolic vulnerability [12]. So, although there are several inhibitors of key cancer enzymes with astonishing results in vitro, and even in preclinical in vivo models, their benefits are more limited when translated into clinical trials either by lack of efficacy or toxicity. Some explanations for such lack of efficacy include similarities between cancer and immune cells metabolism [13,14], the supporting metabolic activities of cancer-associated cells [15,16,17], and our yet far-from-being-complete knowledge about the actual metabolic and non-metabolic conditionings of the TME. Apart from all this, cancer cells can directly or indirectly, through their influence on cancer-surrounding cells, produce immunosuppressive metabolites that restrict the access or function of antitumor immune cells in the TME. These combined metabolic interactions may enable cancer cells to cover their metabolic needs and to evade immune surveillance and destruction [18,19]. Targeting altered cancer metabolism as a therapeutic avenue is not a new idea. In fact, the use of antifolates to treat leukemia dates back to the 1940s [20]. However, our increasing knowledge about the specificity of cancer-related metabolic alterations and findings showing that not only cancer cells metabolic reprogramming, but also their metabolic interplay with host cells plays an active role in supporting cancer, have widened the range of new therapeutic opportunities.

The assessment of dysregulated intratumor metabolism has long since been exploited for cancer diagnosis, monitoring and therapy. Of the different in vivo imaging techniques that rely on metabolic determinations, positron emission tomography (PET), magnetic resonance spectroscopy (MRS) and magnetic resonance imaging (MRI) stand out as the most widely used. These techniques enable the characterization of discrete metabolic features and constraints of a tumor (i.e., preferred use of a particular nutrient or production of a given immunosuppressive metabolite) and, thus, these techniques can also tailor the therapeutic approach that will best work to target a specific metabolic landscape defined by tumor idiosyncrasy [4,21]. Interestingly, tumor- and TME-derived metabolites can also be found in different biospecimens such as blood, which presents a new biomarker discovery strategy. The metabolomic analysis, understood as the holistic determination of all small molecules present in a biological sample, is a powerful technology to both identify new biomarkers and cancer drivers of tumorigenesis and to provide new mechanistic insights into cancer metabolism and metabolic crosstalk between cancer cells and host cells in the TME [11,22,23]. By understanding the metabolic characteristics and constraints of tumors in a personalized manner, metabolic-based cancer treatments can be delivered more efficiently and safely.

Here we review how tumor metabolism and its crosstalk with the TME play an important role in supporting cancer initiation and growth, and in modulating the immune response, which collectively pose new therapeutic opportunities. We also provide an overview of current pharmacological approaches targeting such dysregulated metabolic landscapes. Finally, we describe up-to-date metabolite-based technologies that are routinely used in oncology (PET, MRS, MRI, etc.) along with other emerging tools, such as metabolomics, to focus on their capabilities for identifying metabolic targets, discovering biomarkers and monitoring metabolic therapeutic interventions.

## 2. Metabolites Used as Biomass and Energy Sources by Cancer Cells

Oncogenic transformation integrates internal and external signals to orchestrate a metabolic program that supports cell survival, proliferation and dissemination. On the one hand, cancer cells can more efficiently uptake or metabolize common nutrients such as glucose, glutamine or FA. On the other hand, they can adapt to the nutrient and oxygen limitations of the TME using alternative carbon and nitrogen sources [3,4,5,8,11] (Figure 1). The metabolic alterations and vulnerabilities of cancer cells depend on oncogenic mutation, tissue of origin and tissue location [9,10], but also vary over time and in response to therapy [4].

Glucose is the preferred nutrient of cancer cells by definition since the early observations of Otto Warburg in the 1920s [1]. Several attempts have been made to target glucose metabolism in cancer. 2-deoxy-D-glucose (2-DG) is a non-metabolizable glucose analog that can be phosphorylated by hexokinase in the first glycolysis step, but it cannot be further metabolized. The effects of 2-DG go beyond the inhibition of glucose-dependent reactions, and it also interferes with N-linked glycosylation [24]. Despite its efficacy in vitro and in preclinical models, its use as a single agent in a clinical setting does not provide acceptable results, such as lack of efficacy, toxicity and off-target effects [24,25,26]. It has been proposed that through ATP depletion, 2-DG can be used as sensitizing agent in combination with other drugs or with radiotherapy, although contradictory results have been obtained [24,25,26]. An alternative strategy to target glucose metabolism is to inhibit glucose transporters, particularly GLUT1, which is overexpressed in many cancers. GLUT1-specific inhibitor BAY-876 [27] and more promiscuous inhibitors STF-31 [28], WZB117 [29], and Glutor [30], have demonstrated efficacy inhibiting glucose uptake in not only in vitro models, but also in vivo across a wide range of high glycolytic tumors, although none is being tested in clinical trials [28,29,31]. Alternatively, particular glucose metabolism enzymes/transporters can be targeted for therapeutic purposes. The most predominant targets are those in glycolysis, including monocarboxylate transporters (MCTs) and lactate dehydrogenase (LDH). Glucose catabolism via glycolysis produces two pyruvate molecules, with the associated production of two ATP and two NADH molecules. NAD^+^ regeneration is essential to sustain glycolysis, and also for the biosynthesis of the amino acids and nucleotides required for proliferation [32,33,34]. NAD^+^ regeneration can be achieved by respiration in mitochondria, but also by the conversion of pyruvate into lactate catalyzed by LDH. When the NAD^+^ demand to support oxidation reactions exceeds the ATP turnover rate in cells, NAD^+^ regeneration by mitochondrial respiration is limited, which promotes fermentation despite available oxygen [35]. MCTs mediate the transport of both pyruvate and lactate across membranes, and allow not only the release of lactate when produced in excess, but also its import to be used as fuel for TCA oxidation. Both MCT (mainly MCT1) and LDH inhibitors increase intracellular NADH and effectively inhibit glycolysis. MCT1 inhibitors SR13800 (AR-C122982) [36], AR-C155858 (SR138001) [37], BAY-8002 [38] and AZD3965 [39] effectively increase intracellular tumor lactate concentrations which in turn inhibit proliferation in vitro and in vivo [38,39,40,41]. AZD3965 is an orally bioavailable MCT1 inhibitor that is currently in a phase I clinical trial. Increased MCT4 expression is a mechanism of resistance to MCT1 inhibition and a dual MCT1/4 inhibitor, syrosingopine, has been described [42]. LDH inhibitors FX11 [43], NCI-006 and NCI-737 [44] also have the proven ability to inhibit cell proliferation and control tumor growth in vivo [43,45,46]. Although both LDH and MCT inhibitors achieve tumor control, increased mitochondrial catabolism compensates for deficient glycolysis and, thus, only the glycolysis and respiration (OXPHOS) inhibition combination achieves efficient tumor regression [42,45,47]. Like MCT and LDH inhibitors, OXPHOS inhibitors increase intracellular NADH. Increased mitochondrial NADH shuts down TCA and induces both an energy and a biosynthesis crisis [48]. The drugs targeting mitochondrial complex I, such as IACS-010759 [49], or antidiabetic drugs metformin and phenformin [50,51], effectively inhibit growth in those tumors that rely on OXPHOS for energy production or depend on TCA for aspartate biosynthesis [49,52,53,54].

Glutamine is the most abundant circulating amino acid in blood and plays a key role in energy production, redox homeostasis, macromolecular synthesis and signaling in cancer cells [55,56]. Thus, glutamine metabolism inhibition has been approached as a therapeutic target in cancer for several decades. 6-diazo-5-oxo-L-norleucine (DON) is a broad glutamine antagonist with potent in vitro efficacy. However, its unacceptable toxicity hampers its clinical application [57]. To minimize DON side effects, a series of prodrugs of DON have been designed to circulate and remain intact until they arrive at the TME for their in situ enzymatic cleavage to release the drug [57,58]. Of them, [ethyl 2-(2-amino-4-methylpentanamido)-DON] (also known as JHU083 and DRP-083) has demonstrated glutamine metabolism inhibition in a wide array of tumors in preclinical in vivo models [59]. An alternative strategy for the complete glutamine metabolism blockade is to inhibit glutaminase (GLS), which is a key enzyme in anaplerotic glutamine use [55,56]. The efficacy of the small-molecule inhibitor of GLS, including compound 968, BPTES and its derivative CB-839 in vitro and in preclinical in vivo models, offers a new possibility of targeting glutamine metabolism in cancer [60,61,62,63,64]. Glutamine catabolism inhibition not only suppresses the oxidative and glycolytic metabolism of cancer cells, but also restores CD8^+^ T-cell function in vivo [59]. The combination of glutamine catabolism with adoptive T-cell transfer or anti-PD1 treatment is superior than any the treatments alone [59,65]. CB-839 (under the name of Telaglenastat) is currently under study in several clinical trials. A completely different strategy to target glutamine metabolism is to block its cellular import. Of the several transporters that can mediate glutamine uptake across the cellular membrane, ASCT2 (encoded by gene *SLC1A5*) seems to play a key role in various cancers [66,67,68]. Blockade of ASCT2 with small-molecule inhibitor V-9302 results in attenuated cancer cell growth and proliferation, increased cell death and oxidative stress, which collectively contribute to antitumor responses in vitro and in vivo [69].

FAs are key nutrients that are not only used as fuel for energy production, but also for biosynthesis of complex lipids. In addition to their metabolic role, FAs play many roles in both cancer cells and the TME [70]. Uptake and use of external FAs are particularly relevant for certain cancer types, including ovarian [16,71], colorectal [72], breast [73,74] and mutant KRAS lung cancer [75], while FA translocase CD36 has been demonstrated to play a key role in metastasis [71,76]. CD36 inhibition using either anti-CD36 monoclonal antibodies (mAbs) or oleic acid analog sulfo-n-succinimidyl oleate (SSO), which irreversibly binds CD36, have proven efficacy in preventing tumor growth and metastasis in preclinical models [71,76,77,78]. However, CD36 is relevant not only to provide FA in cancer cells, but also sustains tumor-supportive MDSCs, Tregs and M2 macrophages [79,80,81,82]. Targeting Tregs with a mAb against CD36 elicits additive antitumor responses with anti-PD1 immunotherapy [79]. FA oxidation (FAO) can be halted by the inhibition of carnitine palmitoyltransferase I (CPT1), which enables FAs to enter mitochondria through their binding to carnitine, which is the rate-limiting step in mitochondrial FAO. Etomoxir is an irreversible CPT1 inhibitor whose value in preclinical models for preventing tumor growth and metastasis has been demonstrated [73,83,84]. However, recent studies have shown that the actual antitumor effects observed at the high doses required to inhibit tumor growth might be due to the impairment of OXPHOS rather than actual CPT1 inhibition [85,86,87]. Given the role of enhanced FA uptake by cancer and cancer-associated cells in promoting tumorigenesis through immune suppression [88,89], targeting lipid metabolism holds an very high potential for the development of new treatments to synergize with current immunotherapies [70].

One key ability of cancer cells is their adaptation to the dynamic environmental conditions of the TME, including the use of alternative carbon and nitrogen sources. The enzyme acetyl-CoA synthetase 2 (ACSS2) is expressed in a wide range of cancer cells and allows the use of acetate to generate acetyl-CoA that is used, in turn, for FA synthesis, protein acetylation and energy production [90,91,92]. *ACSS2* expression is up-regulated under metabolically stressed conditions (low oxygen and low nutrient/lipid availability) and *ACSS2* silencing has been shown to reduce the growth of tumor xenografts [92]. Small-molecule inhibitors VY-3-135 and VY-3-249 impair tumor growth in vivo in a breast cancer model that shows high *ACSS2* expression [93] and in a model of obesity-induced myeloma [94], respectively. The development of small-molecule inhibitors of ACSS2 is an active research field [95]. In proliferating cells, glucose and glutamine are not the sources of the majority of cell mass, and non-glutamine amino acids provide abundant carbon and nitrogen for biomass and also for energy production [96]. Some cancer types show increased dependence on BCAAs for protein synthesis, carbon and nitrogen sources and for energy production. Catabolism of BCAAs is mediated by BCAA aminotransferase 1/2 (BCAT1/2). The knockdown or pharmacological inhibition of BCAT1/2 results in decreased proliferation and tumor growth of BCAT1/2-dependent cancer cells [97,98,99,100]. Thus, BCAT1/2 inhibition is a promising therapeutic target in a subset of cancers. Ammonia is a ubiquitous by-product of cellular metabolism. It has been recently demonstrated that ammonia in mice accumulates in the TME and is used by breast cancer cells directly to generate amino acids through GDH activity. Thus, the recycling of circulating ammonia can support cancer biomass and can be pharmacologically exploited to treat cancer [101].

## 3. Oncometabolites

Oncometabolites can be defined as metabolites whose abnormal accumulation causes both metabolic and non-metabolic dysregulation and potential transformation to malignancy [102]. To date, three oncometabolites have been identified: fumarate, succinate and D-2HG. The accumulation of fumarate and succinate results from loss-of-function mutations in mitochondrial Krebs cycle enzymes FH and SDH, respectively. D-2HG accumulation is the result of a gain-of-function in either IDH1 or 2, respectively localized in the cytoplasm and mitochondria. Wild-type (wt) IDH1/2 homodimers catalyze the NADP^+^-dependent and reversible conversion of isocitrate into α-ketoglutarate (α-KG), whereas the heterodimers between mutant and wtIDH1/2 display neomorphic activity that allows the reduction of α-KG directly to D-2HG in the presence of NADPH [102,103,104] (Figure 2).

Succinate, fumarate and D-2HG have individual and shared mechanisms of action. A common oncogenic mechanism linking D-2HG, succinate and fumarate is the inhibition of α-KG–dependent dioxygenases, which results in epigenetic alterations that impede normal differentiation programs and, thus, induce transformation [102,105]. Succinate and fumarate also inhibit α-KG-dependent prolyl-hydroxylase (PHD), which creates a pseudohypoxia state through hypoxia-inducible factor 1α (HIF1α) stabilization. In addition to this, fumarate can modify proteins by succination, a post-translation modification of cysteine residues by forming S-(2-succino)-cysteine. Fumarate accumulation upon FH loss-of-function mutations induces the succination of several proteins, including aconitase and Kelch-like ECH-associated protein 1 (KEAP1). KEAP1 succination disrupts its interaction with transcription factor nuclear factor erythroid 2-related factor 2 (NRF2) which, thus, prevents its proteosomal degradation. Increased NFR2 activity promotes a reductive environment that supports survival and proliferation [102,105].

Inhibition of mIDH1 or mIDH2 with small-molecule inhibitors specific of mutant isoforms lowers D-2HG levels and induces differentiation in vitro and in vivo to result in prolonged survival in preclinical models [106,107,108]. mIDH1 inhibitor Ivosidenib (AG-120) [109,110] and miDH2 inhibitor Enasidenib (AG-221/CC-90007) [111] have been approved by the FDA to treat acute myeloid leukemia (AML) and there are several candidates in clinical trials, including LY3410738, Olutasidenib, BAY 1436032 or IDH305 [112]. One ongoing development is to improve brain penetrance, which is key to extend the application of mIDH1/2 inhibitors to gliomas, and a first-in-class brain-penetrant dual inhibitor of mIDH1 and mIDH2, Vorasidenib (AG-881), is currently being evaluated in clinical trials to treat glioma [113]. The commonest IDH1 mutation in gliomas affects codon 132 and encodes IDH1(R132H). From an immunological perspective, IDH1(R132H) represents a potential immunotherapy target because it is a tumor-specific potential neoantigen that is indeed presented in major histocompatibility complex class II (MHCII) and can, thus, evoke a robust immune response in CD4^+^ T cells [114]. The first-in-humans phase I trial of an IDH1(R132H)-specific peptide vaccine has been recently reported [115].

## 4. Immunosuppressive Metabolites and Metabolic Activities

Cancer cells adapt their metabolism to grow and survive in a hostile environment. This rewired metabolism promotes other effects that sustain tumor progression through immune escape. The preferential use of nutrients by cancer cells or other cell types in the TME (cancer-associated fibroblasts (CAFs), myeloid-derived suppressor cells (MDSCs), tumor-associated macrophages (TAMs), etc.) may restrict their uptake by immune cells and, thus, compromise their functionality. Additionally, in the TME, immunosuppressing metabolites can be released by cancer cells and other resident cells to promote cancer cells immune escape.

It has been shown in highly glycolytic tumors that glucose depletion in tumor interstitial fluid (TIF) leads to T-cell impairment and, hence, inadequate immune tumor suppression [89,116,117,118]. However, glucose depletion might not be the only, or even the commonest, nutrient deprivation that immune cells encounter in the TME. It has been recently reported that tumor-infiltrating T lymphocytes (TILs) display a similar per cell glucose uptake as cancer cells [119]. Indeed, the cells displaying the highest per cell glucose consumption are TAMs and MDSCs, while cancer cells display the highest per cell consumption rate of FAs and glutamine [119]. Thus, while glucose restriction in the TME might occur in some circumstances, deprivation may affect other metabolites. Indeed, in situations in which the reported glucose levels in TIF come close to those encountered in plasma, a sharp drop in other key nutrients, including arginine, tryptophan and cystine, has been found [120]. Although cancer cells themselves modify the metabolic composition of TIF, tumor-supporting cells in the TME play a key role in this sense. For instance, MDSCs and TAMs are responsible for the depletion of amino acids arginine, tryptophan and cystine from TIF [18,121]. The involved mechanisms include arginine degradation via arginase 1 (ARG1) [122], by sequestering cystine and limiting cysteine availability [123] and by catabolizing tryptophan via indoleamine 2,3 dioxygenase 1 (IDO1) [124,125]. Some cancer cells can also degrade tryptophan through an alternative pathway by tryptophan-2,3-dioxygenase 2 (TDO2) activity [126,127]. In addition to the depletion of those nutrients required for sustaining proper immune surveillance of tumors, tumor and tumor-associated cells also produce metabolites that directly inhibit the immune cell function, including kynurenine, D-2HG, adenosine, lactate, 1-methylnicotinamide (1-MNA) and methylglyoxal (Figure 3). The concomitant use of immunotherapy and complementary therapies to diminish the immunosuppressive metabolic properties of the TME is promising for improving the benefit of immune-based therapies.

L-Arginine is required to sustain T-cell growth and proliferation [128]. Myeloid cells in the TME contribute to immune suppression by depleting arginine via increased ARG1 activity [122]. ARG1 inhibition with small-molecule inhibitor CB-1158 blocks the myeloid cell-mediated suppression of T-cell proliferation in vitro and reduces tumor growth in many mouse models of cancer as a single agent and also in combination with checkpoint blockade, adoptive T-cell therapy, adoptive natural killer (NK) cell therapy and chemotherapy agent gemcitabine [129].

Both tumor cells and TAMs/MDSCs can decrease tryptophan availability in the TIF to levels that compromise the immune function [124,125,126,127]. Additionally, tryptophan catabolism through IDO1/TDO2 produces kynurenine, an endogenous ligand of human aryl hydrocarbon receptor (AHR), which limits effector T-cell proliferation [127], promotes the differentiation of CD4^+^ T cells into immunosuppressive regulatory T cells (Tregs) [130,131], and negatively regulates dendritic cell immunogenicity [132]. Although the inhibition of IDO1 and TDO2 alone does not lead to satisfactory results, it enhances tumor regression and control when combined with immunotherapy [126,133,134,135,136]. Currently, several IDO inhibitors (Indoximod, Epacadostat, Navoximod, Linrodostat Mesilate, PF-06840003, KHK2455, etc.) form part of clinical trials to evaluate their efficacy in enhancing immunotherapy in various cancer types [137]. Although it has been recently announced that Epacadostat plus pembrolizumab (anti-PD-1) does not improve progression-free survival or overall survival compared to placebo plus pembrolizumab in patients with unresectable or metastatic melanoma [138], the reasons for failure are unclear. Ongoing clinical trials report encouraging preliminary results, and the usefulness of IDO1 inhibitors as a strategy to enhance anti-PD-1 therapy activity in cancer is still under study. A different approach to target the tryptophan/kynurenine inhibitory axis is to inhibit signaling through AHR. Administering an AHR inhibitor in combination with immunotherapy (anti-PDL1) significantly prolongs the survival of mice bearing mIDH1 tumors, but not in those bearing wtIDH1 tumors, compared to immunotherapy alone. Although the AHR inhibitor can directly affect the signaling of kynurenine in T cells, it seems that the main effect in this case is exerted on macrophages [139].

In addition to its capacity to induce malignant transformation, oncometabolite D-2HG also has immunosuppressive properties. Compared to gliomas with wtIDH1/2, IDH-mutated gliomas show small numbers of tumor-infiltrating CD4^+^ and CD8^+^ T cells [140] and a low expression of cytotoxic T-lymphocyte–associated genes and IFN-γ–inducible chemokines, including CXCL10 [141]. D-2HG can be taken up by T cells, which interferes with the calcium-dependent transcriptional activity of the nuclear factor of activated T cells (NFAT) by inhibiting ATP-dependent TCR signaling and polyamine biosynthesis in T cells. Thus, D-2HG has paracrine effects on T cells that suppresses T-cell activation, proliferation and cytolytic activity, and compromises antitumor immunity [142]. It has been recently reported that tryptophan catabolism is a key modulator of immune suppression in mutant IDH1 (mIDH1) gliomas. D-2HG accumulation in the TME blocks the differentiation of infiltrating macrophages and induces a re-orchestration of tryptophan metabolism, including increased AHR expression, TDO2 activity and kynurenine production [139]. Due to the immune suppression properties of D-2HG, the antitumor properties of immunotherapies based on a mIDH1-specific vaccine or checkpoint inhibition are improved by inhibition of mIDH1 in tumors bearing that particular mutation [141,142]. 

Increased adenosine levels have been reported in the TME [143,144]. Adenosine and associated nucleotides (i.e., ATP, ADP, AMP) can be passively released as a result of cancer tissue hypoxia, ischemia and necrosis, but also actively due to altered metabolism in cancer and cancer-associated cells [145]. Adenine-containing nucleotides can be hydrolyzed in the TME by the sequential activities of ectoenzymes CD39 and CD73, which convert ATP (or ADP) into AMP and AMP into adenosine, respectively [145]. They are present in both cancer and immunosuppressive cells in the TME [146,147,148,149,150]. Adenosine signaling through the interaction with adenosine receptor A2A impairs NK [151] and effector T cells [152,153,154], and promotes the generation of immunosuppressive Tregs [155]. In preclinical models, the targeted inhibition of the immunosuppressive adenosine pathway by targeting CD73, CD39, or adenosine receptors A2A or A2B with either small-molecule inhibitors or mAbs can restore antitumor immunity and enhance the efficacy of cancer immunotherapies [156,157,158,159,160,161]. To date, several mAbs against CD73 (including oleclumab (MEDI9447), BMS-986179, CPI-006 and NZV930), mAbs against CD39 (TTX-30, SRF617 and IPH5201), small-molecule inhibitors against CD73 (AB680, CB-708 and LY3475070) and small-molecule A2A receptor antagonist (AZD4635, taminadenant (NIR178; PBF-509), ciforadenant (CPI-444), EOS100850 and preladenant (MK-3814A)), as well as one dual A2A and A2B antagonist (AB928), have entered into clinical trials. Promising results are reported for those that are already being tested, usually in combination with immunotherapy [162].

Lactate is the end product of glycolysis. Consistent with the Warburg effect, lactate is the most consistently up-regulated metabolite across diverse tumors [11]. In vitro, high concentrations of lactate/lactic acid can drive T cells toward an immunosuppressive Treg phenotype [163] and limit the proliferation and antitumor function of both innate and adaptive lymphocytes [163,164,165]. In vivo, high *LDHA* expression is associated with poor immune surveillance in the TME [165,166]. However, while decreased glucose levels in the TIF have been reported in highly glycolytic tumors, corresponding lactate data are in most cases not available or do not show an increase up to levels that impede proper immune response [11]. Conversely, in highly glycolytic tumors or in tumors with high *LDHA* expression, lactate may accumulate inside tumor cells and the immunosuppressive effects can be due to acidic pH and insufficient glucose levels (among other required nutrients) in the TIF [11]. Nevertheless, preventing lactate production (i.e., via glycolysis inhibition with MCT or LDH inhibitors) is a valid strategy to avoid tumor growth and restore immune function as previously discussed. Moreover, transient LDH inhibition enhances the generation of memory CD8^+^ T cells capable of triggering robust antitumor responses after adoptive transfer [167].

Nicotinamide N-methyltransferase (NNMT), which catalyzes the transfer of a methyl group from S-adenosylmethionine (SAM) to nicotinamide, is overexpressed in a variety of human cancers (in both cancer and cancer-supporting cells), where it contributes to tumorigenesis through epigenetic remodeling by creating a metabolic methylation sink [168,169]. The product of the reaction, 1-MNA, can be taken up by TILs. Functionally, 1-MNA induces T cells to secrete tumor-promoting cytokine tumor necrosis factor alpha (TNF-α), reduces IFN-γ secretion in CD4^+^ T cells and has been shown to induce a significant reduction in chimeric antigen receptor (CAR)-T-cell killing activity [170]. The development of potent specific NNMT inhibitors with favorable pharmacokinetic and pharmacodynamics properties is an active, yet still emerging research field that may induce a favorable response in combination with immunotherapy through tumor growth restriction and a decrease in immunosuppressive metabolite 1-MNA [171,172,173,174].

We have already discussed the key role of MDSCs in the metabolic and non-metabolic immune suppression in the TME. It has been recently shown that production of the glycolysis by-product methylglyoxal by the enzyme semicarbazide-sensitive amine oxidase (SSAO) can be an additional metabolic mechanism by which MDSCs inhibit T cells. Methylglyoxal is a reactive compound that attacks amino/guanidine-groups (HN = C–(NH2)–NH) and, thus, preferentially targets amino acids l-lysine and l-arginine, as well as their residues in proteins, to form advanced glycation end products (AGPs) that can render amino acids and proteins non-functional. Upon the transfer to T cells, methylglyoxal depletes arginine and causes a concomitant increase in the methylglyoxal-derived glycation products of l-arginine. As a result, methylglyoxal completely halts T-cell activation, proliferation and cytokine production [175]. In a murine cancer model, neutralization of methylglyoxal with dimethylbiguanide released MDSC-mediated T-cell suppression and, together with checkpoint inhibition, improved the efficacy of a vaccine-based cancer immunotherapy [175].

## 5. Noninvasive Measurement of Metabolites as Biomarkers in Cancer

We have shown how altered TME metabolism can be targeted for therapeutic purposes. Additionally, such cancer metabolism can be also informative of tumor presence or tumor status. Therefore, the noninvasive assessment of these tumor-derived metabolites could appear as a strategy to discover new biomarkers (diagnosis, prognosis, monitoring, stratification, etc.), which would be very interesting from a clinical perspective. Most of the above-mentioned metabolic alterations have been discovered by either using in vitro models or analyzing tumor biopsies from patients and animal models. However, they can be assessed in patients by two complementary approaches. The in vivo metabolism assessment in tissues can be performed by MRS, PET-based and MRI techniques [21]. Additionally, the altered metabolic properties of tumors may result in changes of the circulating and/or excreted metabolites present in readily accessible biological samples. There are different techniques that allow the sensitive and accurate determination of small molecules (i.e., metabolites) in different biological fluids, including enzymatic assays (coupled to various detection methods, such as absorbance, fluorescence or electric current/potential), liquid chromatography (LC) or gas chromatography (GC) separation usually coupled to mass spectrometry (MS) detection and nuclear magnetic resonance (NMR). However, to date, LC-MS stands out as the most widely used technique for both exploratory and validated targeted analyses [22].

### 5.1. MRS/MRI/PET

PET, MRS and MRI are complementary techniques that allow the study in vivo of intact tissues. MRS uses high-powered magnetic fields to quantify chemical compounds in tissue. Endogenous isotopes detectable by magnetic resonance include ^1^H (protons) and ^31^P, whereas ^13^C, ^19^F, and ^17^O containing tracers can also be used. PET measures the distribution of positron-emitting radioactive isotopes from an injected tracer. These radioisotopes include ^11^C, ^13^N, ^15^O, and ^18^F. In MRI, the focus is placed on the anatomical characterization of a given nucleus [21]. ^18^F-2-deoxyglucose (18FDG) is a glucose analog that can be imported and retained (phosphorylated) by cells, but not further metabolized. In addition, it contains a positron-emitting radioactive isotope (^18^F) that can be detected using PET. Thus, upon intravenous 18FDG infusion in patients, tumors can be localized by PET-MRI thanks to their increased capacity to accumulate the tracer [176]. Pharmacological GLUT1 inhibition can be monitored using 18FDG, and it results in a decreased signal [28] (Figure 4A). However, 18FDG is ineffective in evaluating tumors such as gliomas because of high background uptake in the brain, or in tumors/tumor regions where the increase in glucose metabolism is not that acute. Of the wide-ranging PET tracers, 4-^18^F-(2S,4R)-fluoroglutamine (18FGln) and 1-^11^C-acetate (11CAc) are particularly interesting [177,178,179]. In addition to their role as alternative PET tracers for tumor imaging, they can be used to point to more sensitive tumors to the inhibition of glutamine metabolism and ACSS2, respectively. In response to the pharmacological inhibition of glutamine uptake, the intratumoral accumulation of 18FGln reduces [69,177,180], whereas the intratumoral levels of 18FGln rise upon GLS inhibition [181] (Figure 4B,C).

MRS can detect metabolites whose concentration is above 1mM. So only a limited subset of metabolites can be quantified in vivo using MRS. Although altered levels of some metabolites have been proposed as cancer biomarkers, to date 2HG accumulation in patients with mIDH1/2 gliomas is the only one to have shown its actual value as a noninvasive cancer-associated biomarker using MRS [21,182,183,184,185] (Figure 4D). The utility of MRS is limited by its low sensitivity, while the main drawback with ^18^F-based PET tracers is associated with loss of ^18^F. Thus, ^18^F-based PET tracers allow detection of the accumulation of the infused tracers, but not to study actual metabolism in vivo. Using hyperpolarized ^13^C tracers results in an enhancement of the signal related to conventional MRI, while allowing measurement of the downstream products of the injected tracer [186]. We previously mentioned that many tumors show high MCT and LDH levels to sustain glycolysis. Hyperpolarized 1-^13^C-pyruvate can be taken up by cells and converted into 1-^13^C-lactate. Upon hyperpolarized 1-^13^C-pyruvate infusion in patients with prostate cancer, biopsy-proven cancerous regions show increased hyperpolarized 1-^13^C-lactate levels and a high 1-^13^C-lactate/1-^13^C-pyruvate ratio [187]. In preclinical models, the 1-^13^C-lactate/1-^13^C-pyruvate ratio lowers upon chemotherapy [188] but, more interestingly, they can be used to identify tumors susceptible to LDH or MCT inhibition, and to monitor in vivo the response to targeted therapies against those key glycolysis-related proteins [45,189,190] (Figure 4E). The direct measurement of the glycolysis pathway can be done instead using hyperpolarized U-^2^H, U-^13^C-glucose, which results in ^13^C labeling in lactate, but also in other glucose products as the intermediate on the oxidative branch of the pentose phosphate pathway 6-phosphogluconate. Increased labeling in lactate is observed in tumors compared to adjacent normal tissue, and the signal decreases upon treatment with chemotherapy [191]. In addition to the direct 2HG measurement in gliomas with mIDH1/2, the in vivo production of 2HG can be measured using hyperpolarized 1-^13^C-α-KG [192] or 1-^13^C-glutamine [193]. Upon the pharmacological inhibition of mIDH1, glutamine-derived 2HG production decreases [193] (Figure 4F).

### 5.2. LC-MS

MS-based metabolomics analyses are typically performed by two complementary approaches known as untargeted and targeted metabolomics. The former aims to provide the widest possible metabolome coverage, while the latter focuses on the quantification of a pre-determined set of known metabolites [22]. In the exploratory analysis, untargeted metabolomics is key to identify altered metabolites. Proper validation and transfer to the clinic requires the development of rapid, specific, and quantitative measurements using targeted methods and different cohorts of patients from those used for biomarker discovery. To exemplify the importance of proper validation when the aim is to translate a biomarker into the clinic, using the untargeted characterization of clinical samples (including tissue, plasma and urine) sarcosine, an N-methyl derivative of glycine, was identified as a differential metabolite that substantially increased during prostate cancer progression to metastasis and was detected non-invasively in urine [194]. However, the attempts to reproduce this observation have failed to validate sarcosine as a biomarker of prostate cancer in either serum or urine samples [195,196]. Conversely, (D-)2HG and IDO1/TDO2-related metabolites tryptophan and kynurenine are two examples of the successful application of LC-MS-based technology to assess metabolites as biomarkers in cancer. 

D-2HG accumulation is the hallmark of tumors harboring mutations in IDH1/2 [103,104]. Albeit to a lesser extent, it can be produced by enzyme D-3-phosphoglycerate dehydrogenase (PHGDH), the first enzyme in the serine biosynthetic pathway, which is genomically amplified in various cancer types [197]. L-2HG can be produced under hypoxia or acidic conditions by enzymes LDH and malate dehydrogenase (MDH) [198,199]. However, to separate D(R) and L(S) enantiomers of 2HG, chiral chromatographic separation (through 2HG derivatization with a chiral agent or using a chiral column) must be performed. The increase in 2HG is reflected at the systemic level, and mutations in IDH1/2 result in higher circulating levels of D- and total 2HG in AML patients. Quantification of D- or total 2HG can be used to discriminate between tumors/cancers harboring wt and mIDH1/2, and the levels of D- or total 2HG lower upon both mIDH1/2 inhibition and tumor/cancer regression [104,108,109,111,200,201,202,203] (Figure 4G).

High IDO1 levels are associated with poor prognosis in several cancer types [204]. Indeed altered serum/plasma levels of tryptophan and/or kynurenine have been found in various cancer types compared to healthy donors. [204,205,206]. However, the dietary influence on both tryptophan and kynurenine circulating levels hinders their usefulness as prognostic or diagnostic markers in cancer [204]. IDO1 inhibition results in low kynurenine and of kynurenine/tryptophan levels, whereas TDO2 inhibition leads to high tryptophan levels and, thus, low kynurenine/tryptophan levels. Thus, the measurement of circulating kynurenine and tryptophan levels is routinely used to assess the efficacy of IDO1 and TDO2 inhibitor efficacy in both preclinical and clinical studies [136,207,208,209,210] (Figure 4H).

In addition to quantifying small molecules, high-resolution MS allows stable isotope tracers to be used to study metabolism [23]. Although the combination of infused isotope tracers with the characterization of the produced metabolites in circulation has not yet been achieved, intraoperative infusion of isotope tracers followed by the comprehensive metabolic characterization of the resected tumor/biopsy by LC-MS-based metabolomics provides the unprecedented characterization of the in vivo metabolism of tumors. This approach has the potential to be extremely useful not only in decision making, but also in identifying new metabolic vulnerabilities [211,212,213,214,215].

## 6. Conclusions

Tumor metabolism and its crosstalk with the TME is an active research field. Metabolic plasticity in tumors allows their adaptation to the challenging TME conditions, and may eventually enable and support metastasis. Advances in molecular biology, genetic tools and metabolomics have greatly extended the knowledge about tumor metabolism both in vitro and in vivo. The characterization of metabolic fluxes and metabolic co-dependences across a wide variety of tumor types, combined with detailed information about the many cell types making up the TME, has allowed the identification of therapeutic targets whose specific aims are cancer and cancer-supporting cells, while relieving the suppression that antitumor cells suffer in the TME. Such new findings have promoted the development of a wide arsenal of antitumor drugs with the potential to be used alone or in combination with immunotherapies (i.e., immune checkpoint inhibitors, vaccines or adoptive transfer of immune cells, e.g., CAR-T cells). To take full advantage of these new developments and to select the treatment that best works for each patient, we should be able to translate these molecular characterization capabilities used in preclinical experimentation to the clinic. To this end, highly informative, yet invasive, techniques should be replaced with noninvasive ones. Recently, the use of liquid biopsy—mainly blood—for analysis of circulating tumor cells (CTCs) and circulating cell-free tumor DNA (ctDNA) has become a valuable tool to guide treatment and to monitor response to therapy [216]. With direct metabolic measurements, we name a few successful examples of MRS, MRI and PET imaging techniques and the LC-MS-based quantification of circulating metabolites (i.e., D-2HG and kynurenine/tryptophan). However, these approaches do not reproduce the complexity required for profound metabolic characterization and each one only informs about a particular or a very narrow set of metabolic activities. Only by understanding how tumor cells alter circulating metabolites concentrations and how cancer metabolism of infused stable isotopic tracers can be reconstructed from circulating metabolites will we be able to fully unleash the potential of metabolic-based determinations and treatments to guide personalized medicine into the clinic. Hence, LC-MS-based-metabolomics alongside other clinical analyses will soon play an increasingly key role.

## Figures and Tables

**Figure 1 cancers-13-03230-f001:**
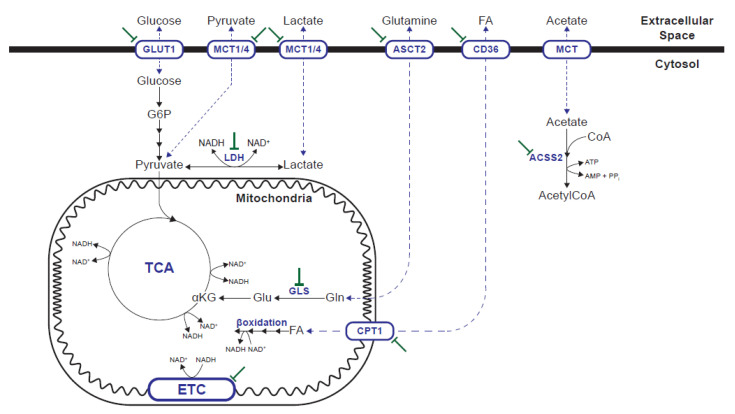
Metabolic pathways supporting biosynthesis and energy production in cancer cells that can be pharmacologically targeted. GLUT1: glucose transporter 1; MCT: monocarboxylate transporter; ASCT2: alanine, serine, cysteine transporter 2, SLC1A5; LDH: lactate dehydrogenase; ACSS2: acetyl-CoA synthetase 2; GLS: glutaminase. CPT1: carnitine palmitoyltransferase 1; ETC: electron transport chain.

**Figure 2 cancers-13-03230-f002:**
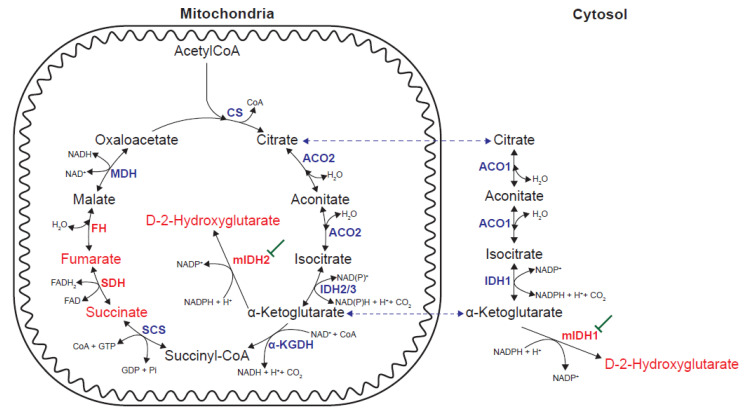
Alterations in enzymatic activities leading to the accumulation of oncometabolites. Gain-of-function mutations in IDH1/2 (mIDH1/2) lead to the production of D-2-hydroxyglutarate. mIDH1/2 can be pharmacologically inhibited. Loss-of-function mutations in SDH and FH lead to the accumulation of succinate and fumarate, respectively. CS: citrate synthase; ACO: aconitase; IDH: isocitrate dehydrogenase; α-KGDH: α-ketoglutarate dehydrogenase; SCS: succinyl-CoA synthetase; SDH: succinate dehydrogenase; FH: fumarate hydratase; MDH: malate dehydrogenase.

**Figure 3 cancers-13-03230-f003:**
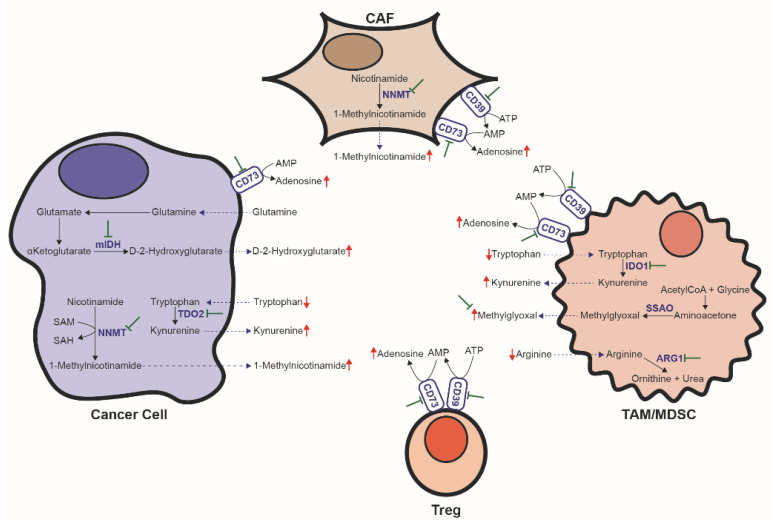
Metabolic activities contributing to immune suppression in the tumor microenvironment. Metabolic activities depleting key nutrients include arginine degradation by ARG1 and tryptophan degradation by IDO1/TDO2. Metabolic activities producing immunosuppressive metabolites include 1-MNA production by NNMT, kynurenine synthesis from tryptophan by IDO1/TDO2, D-2HG synthesis by mIDH1/2, adenosine production from ATP/AMP by the combined activities of CD39 and CD73 and production of methylglyoxal by SSAO. CAF: cancer-associated fibroblast; TAM: tumor-associated macrophage; MDSC: myeloid-derived suppressor cell. mIDH: mutant isocitrate dehydrogenase. TDO2: tryptophan-2,3-dioxygenase 2; NNMT: nicotinamide N-methyltransferase; IDO1: indoleamine 2,3 dioxygenase 1; ARG1: arginase 1; SAM: S-adenosylmethionine; SAH: S-adenosylhomocysteine; SSAO: semicarbazide-sensitive amine oxidase.

**Figure 4 cancers-13-03230-f004:**
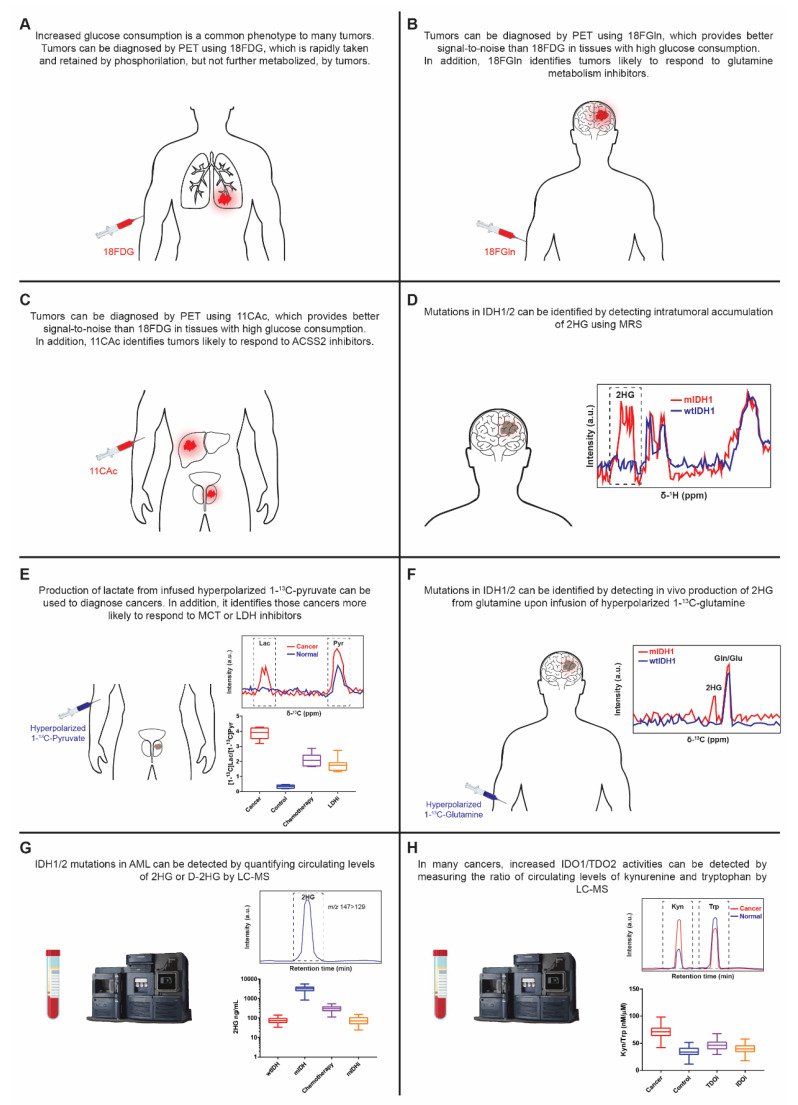
Metabolic activities/metabolites and the associated noninvasive techniques for their assessment with diagnostic, therapy selection and efficacy prospects. (**A**). PET-based detection of infused 18FDG. (**B**). PET-based detection of infused 18FGln. (**C**). PET-based detection of infused 11CAc. (**D**). MRS-based detection of infused 2HG. (**E**). MRI-based detection of 13C-Lactate produced from infused hyperpolarized 1-13C-pyruvate. (**F**). MRI-based detection of 13C-2HG produced from infused hyperpolarized 1-13C-glutamine. (**G**). LC-MS-based detection of circulating (D-)2HG. (**H**). LC-MS-based detection of circulating tryptophan and kynurenine. 18FDG: ^18^F-2-deoxyglucose; 18FGln: 4-^18^F-(2S,4R)-fluoroglutamine; 11CAc: 1-^11^C-acetate; ACSS2: acetyl-CoA synthetase 2; IDO1: indoleamine 2,3 dioxygenase 1; TDO2: tryptophan-2,3-dioxygenase 2.

## Data Availability

No new data were created or analyzed in this study.

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
