# Peer review of "Tumor Microenvironment-Derived Metabolites: A Guide to Find New Metabolic Therapeutic Targets and Biomarkers"

_cancers, 2021, doi:10.3390/cancers13133230_

Round 1

Reviewer 1 Report

In this review, García-Cañaveras and Lahoz have provided a comprehensive overview of tumor microenvironment-derived metabolites as well as the utilization of the metabolites for targeting and non-invasive imaging techniques. As there is increasing interest in uncovering the metabolic changes that occur in the tumor microenvironment, this review will provide the readers with the current knowledge and approaches involving this field. It is overall well-written although there is a need to go through the text to revise multiple typos and I have two minor comments/suggestions for the authors.

First, although it may require some restructuring, I suggest that combining the sections '2. Biologically relevant metabolites in cancer and immune function' and '3. Therapeutic exploitation of metabolic-based cancer alterations' and then dividing them into the sections '1. Metabolites used for cancer energy and biomass sources', '2. Oncometabolites', and '3. Immunosuppressive metabolites'. Because with the current structure, the section '2. Biologically relevant metabolites in cancer and immune function' is completely missing the text explaining glucose, glutamine, or fatty acids and, therefore, can confuse readers by missing this important information. I believe by combining and dividing the sections, it will give the readers a better understanding of this information.

Second, in the case where the sections are divided as mentioned above, Figure 1 can go in section 1 and Figure 2 can go in section 3. Therefore, I suggest that an additional Figure should be made to show the oncometabolites and ways to target these pathways.

Author Response

We thank the reviewer for his/her critical evaluation of our work and for the insightful suggestion about reorganizing the content of the manuscript.

Following his/her recommendation, we have reorganized the manuscript and incorporated a new figure. Now the headings and figures are organized as follows:

  1. Introduction
  2. Metabolites used as biomass and energy sources by cancer cells

Figure 1. Metabolic pathways supporting biosynthesis and energy production in cancer cells that can be pharmacologically targeted.

  1. Oncometabolites

Figure 2. Alterations in enzymatic activities leading to the accumulation of oncometabolites.

  1. Immunosuppressive metabolites and metabolic activities

Figure 3. Metabolic activities contributing to immune suppression in the tumor microenvironment.

  1. Non-invasive measurement of metabolites as biomarkers in cancer

Figure 4. Metabolic activities/metabolites and the associated non-invasive techniques for their assessment with diagnostic, therapy selection and efficacy prospects.

  1. Conclusions

In addition to that, we have thoroughly revised the manuscript and the figures to detect and correct several typos and errors.

Reviewer 2 Report

This is an interesting review article focused on tumour derived metabolites as targets and biomarkers. I have only few minor concerns. 

The minor concerns:

1) The spell check should be performed.

For example:

line 14 should read "required"

lines 20-21 the sentence should be rewritten

line 30 should read "escape"

2) Additional articles should be included.

For example: 

Tumor-Derived Lactic Acid Contributes to the Paucity of Intratumoral ILC2s. Wagner M, Ealey KN, Tetsu H, Kiniwa T, Motomura Y, Moro K, Koyasu S.Cell Rep. 2020 Feb 25;30(8):2743-2757.e5. doi: 10.1016/j.celrep.2020.01.103.

Brand, A., Singer, K., Koehl, G.E., Kolitzus, M., Schoenhammer, G., Thiel, A., Matos, C., Bruss, C., Klobuch, S., Peter, K., et al. (2016). LDHA-Associated Lactic Acid Production Blunts Tumor Immunosurveillance by T and NK Cells. Cell Metab. 24, 657–671.

Author Response

We thank the reviewer his/her positive comments and critical evaluation of our work and for the insightful suggestion on discussing the role of lactate as immunosuppressive metabolite.

Within the section “4. Immunosuppressive metabolites and metabolic activities” we have included a paragraph where we describe the evidence on the role of lactate/lactic acid in inhibiting both adaptive and innate immunity. We have included the references suggested by the reviewer as well as a few extra ones on the topic. While the in vitro evidence is clear about the many ways lactate/lactic acid can limit proliferation and antitumor activity in lymphocytes, the evidence in vivo is not that strong. Increased LDHA activity in cancer cells is associated with decreased immune response in vivo, but the evidence supporting that this effect is directly mediated by the accumulation of lactate/lactic acid in the TIF is absent, and there are other plausible explanations as the decrease in pH and the depletion of glucose in the TIF. Thus, we cite relevant bibliography and provide a critical and reasoned evaluation on the available evidence. The text can be found in lines 404-419.

In addition to that, we have thoroughly revised the manuscript and the figures to detect and correct several typos and errors.